# Continual Learning with Delayed Feedback

## Abstract

Most of the artificial neural networks are using the benefit of labeled datasets whereas in human brain, the learning is often unsupervised. The feedback or a label for a given input or a sensory stimuli is not often available instantly. After some time when brain gets the feedback, it updates its knowledge. That's how brain learns. Moreover, there is no training or testing phase. Human learns continually. This work proposes a model-agnostic continual learning framework which can be used with neural networks as well as decision trees to incorporate continual learning. Specifically, this work investigates how delayed feedback can be handled. In addition, a way to update the Machine Learning models with unlabeled data is proposed. Promising results are received from the experiments done on neural networks and decision trees.

## 1 Introduction

The high representational capacity of neural networks as well as the discovery of good training methods, and the access to fast processing units resulted in super human accuracy in various artificial intelligent tasks. Supervised image classification results has improved to super human levels by various convolutional neural network(CNN) architectures such as AlexNet (Krizhevsky et al., 2012), VGG(Simonyan & Zisserman, 2014), GoogleNet(Szegedy et al., 2015), and ResNetHe et al. (2016). Most of the CNNs are learned with supervision.The limitation of labeled dataset opens the path for unsupervised learning. Moreover, human brain often does continual learning in an unsupervised manner. As an example, take an infant. It does not know most of the things in the environment. But it sees everything. Whenever somebody gives some detail or a label about the object that the baby has seen, it updates its knowledge. This work names this detail or label as delayed feedback. Hence, the key difference in CNNs is that we feed everything with labels at the start which does not occur in human brain. In addition, humans learn continually. The distinction of training and testing is not realized in actual learning of human.

Figure 1 explains a delayed feedback scenario. Suppose an infant sees 3 different cars. But it does not know anything about the car. After sometime, somebody shows a new car and labels it as a car. Now the infant learns that the previous unknowns vehicles should be cars. In this case, the baby does not know any car before seeing these objects. This scenario can be extended to different types or shapes of cars as well where baby knows about one set of cars and how it can learn other different cars.

Unsupervised learning is discussed predominantly in 2 categories. One is reconstructing the original data using auto-encoders and the latent representations are used for the downstream tasks(Hinton et al., 2011). Especially this resulted in good dimensionality reduction techniques such as t-SNE (Maaten & Hinton, 2008). The second one is instead of reconstructing the given data, the model learns a contrastive task which can produce latent representations which are useful for other AI tasks (Hjelm et al., 2018), (Schneider et al., 2019). Both categories can find better representations to capture the variability of the data in the input space. In either ways, the task specific information extraction is not possible due to learning the same data or a contrastive task. The area of updating a supervised model parameters using an unlabeled data is not explored as per our knowledge. In this work, an idea is proposed to update the model to improve the accuracy over some new data. Mainly, two contributions are presented in this paper.

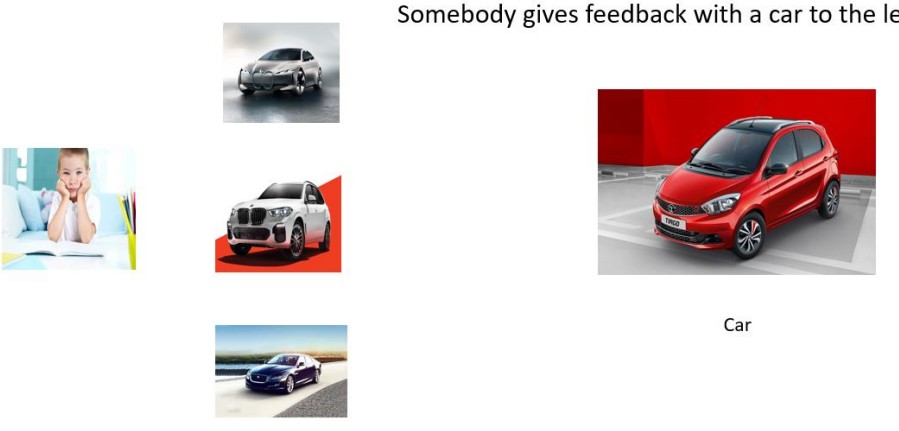

Figure 1: Delayed Feedback

1. A way to handle delayed feedback is presented and implemented by maintaining a queue to store some recent data of samples. This concept introduces the same delayed feedback of actual brain in artificial neural networks.

2. Instead of having training and testing phase, a new continual learning model is proposed which can update the model even with unlabeled data.

## 2 RELATED WORK

### 2.1 CONTINUAL LEARNING

Neuroscience literature has widely spoken about continual learning ((Hassabis et al., 2017) , Parisi et al. (2019)). However, the literature on continual learning in artificial neural network is not studied deeply and still at the inception level. Continual learning is mainly discussed via resolving the issue of Catastrophic Forgetting ((Kirkpatrick et al., 2017)). Catastrophic forgetting occurs when a neural network forgets most part of the previously learnt task while learning a new task. The distributional shift in inputs is not clearly modeled using neural networks. Hence, neural networks perform badly with the old tasks. Superposition of models into a single neural network ((Cheung et al., 2019)) and neural network pruning ((Golkar et al., 2019)) are proposed to do continual learning.

### 2.2 DELAYED FEEDBACK

Delayed Feedback is an important phenomenon in human learning. Brain encounters different objects routinely without knowing the actual identifier of what it is. When some stimulus gives the label, brain updates the knowledge. If there is no signal for a certain period of time, then the object seen will be erased from memory. This mechanism is named as delayed feedback. Speech and visual delayed feedback is discussed previously ((Lee, 1950), (Smith, 1962)). But the effect of delayed feedback is vastly unexplored in artificial neural networks.

Data hungry supervised algorithms achieved state of the art results in popular AI tasks such as image classification, recognition, and segmentation, but the availability of labeled dataset is always limited than the abundant unlabeled data in the wild. The availability of abundant unlabeled dataset and delayed feedback can jointly build strong continuous learning models.

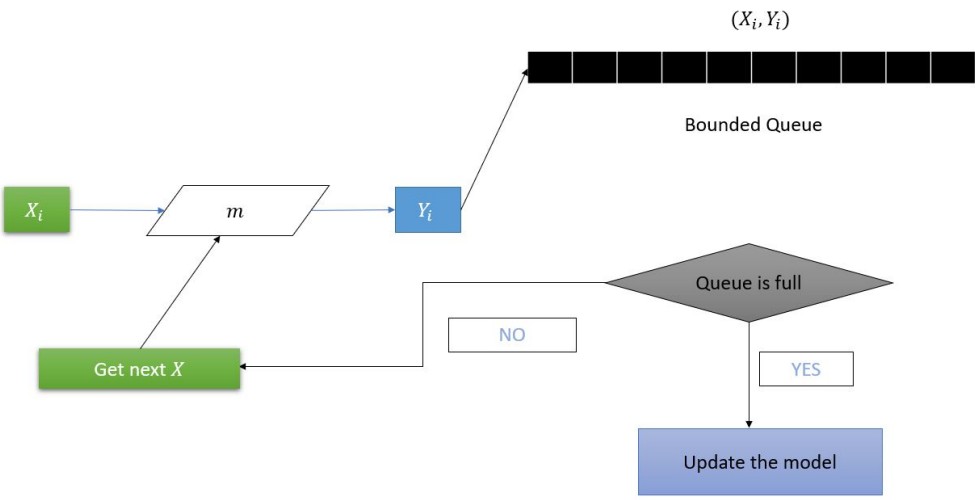

Figure 2: Growing Phase

# 3 CONTINUAL LEARNING FRAMEWORK WHICH HANDLES DELAYED FEEDBACK

## 3.1 BASIC MODEL

This work introduces 2 terms **seeding** and **growing** instead of training and testing. **Seeding** is the phase where the initial learning occurs. This is the phase where the model takes advantage and learns from labeled data. **Growing** is the phase where the model learns information from new unlabeled data and updates the model. The learnt neural network of a classification problem can be seen as a function which produces confidence values($f(X) = Y$). Initial function $f$ is improved to $f'$ in the conventional training of a neural network, same thing happens here with **seeding**. But the function $f$ is fixed in conventional neural networks in testing phase where as in **growing** phase of the continual learning framework the model parameters are updated with the framework rules. Hence mostly a new function($f''$) will be seen after every new inputs.

The figure 2 illustrates how the frame work works in **growing** phase. When a new sample($X_i$) comes in(for an image classification problem, $X_i$ is an image), the model will predict confidence value for the input. Confidence value is the probability vector($Y_i$) for each sample(In a neural network, $Y_i$ is the output from softmax layer). The input($X_i$) and the confidence value vector($Y_i$) are inserted to a bounded queue. If the queue is not full, the next sample will be processed and appended to the queue. When the queue is full, an element from queue is removed based on the following criteria. In addition, one hot encoding is randomly created to give the feedback ($Z_0$).

1. Queue is searched for confidence value($Y_i$) which has the lowest distributional distance or the highest similarity with the one hot encoding($Z_0$). Bhattacharyya coefficient is used to measure the similarity between two distributions.

2. Now the best $Y_1$ is selected to match the given feedback. $Y_1$ has a corresponding image ($X_1$). Now the model is updated with the sample ($X_1, Z_0$). The element ($X_1, Y_1$) is removed from the queue.

3. Assume the learning rate used for **seeding** is $r$. The learning rate used in **growing** phase is $r'$ which is around 10 times smaller than $r$. When we update the model with sample ($X_1, Z_0$), we use a higher learning rate($r' \times s$ where $s > 1$) if the bhattacharyya coefficient between $Z_0$ and $Y_1$ is in $[0.5, \epsilon]$ (Here $\epsilon < 1$). If $y_{max} > \epsilon$, the model is updated with a small learning rate ($r'/s$ where $s > 1$). $\epsilon = 0.8$ used in most of the experiments.

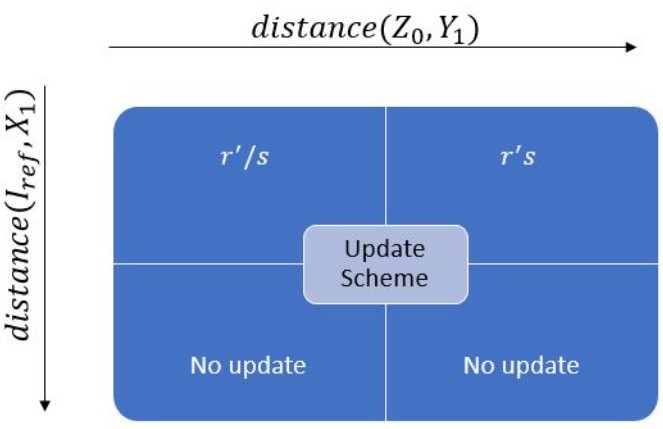

Figure 3: Update rule with CBIR Features

4. If there are no samples with bhattacharrya coefficient $\geq 0.5$, then the oldest element in the queue is removed (dequeue operation in the queue) without any update to the model.

The reason behind selecting a lower learning rate for high confidence vector is that the model is confident about the decision. Hence, updating with a higher learning rate has a higher probability of overfitting or fitting a wrong sample with high confidence. If the confidence values are in $[0.5, \epsilon]$, then the model is not highly confident, but fair enough to say that these decisions will be correct. Hence, we are updating these samples with higher learning rate. Having a good initial model from seeding leads to improvement in accuracy. But starting with a bad model with respect to accuracy may result in models which are worse than initial model. Take the same infant example, if the infant is taught wrong things from the start, then it will make more decisions which are wrong. Similar thing can happen here as well.

Figure 3 illustrates the update rules. If the distance in the input space is higher ($I_{ref}$ and $X_1$), then no update is done. If the distance in the input space is lower, then the same update rule explained in section 3.1 is applied to the model. The update of the model is restricted by the input space similarity. This update rule increased the accuracy by a good margin.

In the experiments, $s = 5$ is used update the model. Selecting a higher $r^{'}$ than $r$ is not desirable because the probability of getting an inaccurate model increases with $r^{'}$s which are greater than $r$.

Seeding phase contains most of the updates to the model. It is seeded with several epochs whereas growing phase deals with a single update for each sample. If the initial learnt function from seeding is $g$, the growing phase functions will fall in a small manifold around $g$.

Continual learning is different from online learning. In online learning, samples come with labels and the update is done instantly. But in continual learning, instant update is not done. Instead of updating the samples are kept in queue. When the queue is full, updates are done as described in this section. The learning is almost fixed in online learning at least for a certain period of time whereas in continual learning, it depends on the sample. Infact, machine learning models which have online version are easy to do continual learning since the updates can be done partially for the given samples.

## 3.2 EXTENSION WITH CONTENT BASED IMAGE RETRIEVAL(CBIR) FEATURES

The feedback is just given as a label, $Z_0$. In addition to the label, a reference image ($I_{ref}$) for label $Z_0$ is passed. Reference images can be collected from initial samples used for **seeding**. When it comes to calculate similarity, the distributional similarity is calculated between $Z_0, Y_1$. More-

over the queue contains the images correspond to all labels. Content based similarity is calculated between $I_{ref}$ and $X_1$. The model update decision is done as shown in figure 3. Content Based Image Retrieval Systems use various features such as color, shape and texture. In this work, color histograms are used to measure the similarity between $I_{ref}$ and $X_1$.

### 3.3 EXTENSION WITH BATCHES

Instead of searching for a single element in the queue, searching for multiple elements and update the model together resulted in better model than a single sample update model described in section 3.1. In an image classification problem with 10 classes, feedback $Z_0$ will be sampled with probability $1/10$ for a class. Suppose if we have 2 samples which are closer match for a generated feedback $Z_0$, then the model can be updated together. Indeed, it resulted in models with better accuracy and reduction in **growing** time. Not doing batch update may keep good samples longer in the queue.

## 4 EXPERIMENTS AND RESULTS

### 4.1 CONVOLUTIONAL NEURAL NETWORKS(CNNS) BASED EXPERIMENTS

#### 4.1.1 EXPERIMENT SET UP

A small CNN with around 66000 parameters and 3 convolutional layers is used for all the evaluations. The CNN used for traditional neural network functionality is noted as T-CNN and the CNN used for continual learning is noted as CL-CNN. But the structure of the network is same in either cases. CIFAR10(Krizhevsky et al., 2009) dataset is used for evaluation. During seeding phase CNN is trained over 200 epochs. Stochastic Gradient Descent is used for updating the weights of CNNs. Learning rate $r = 0.0001$ is used in **seeding**. CNNs are trained/seeded over 8 different initial weights which were generated from seeding different random state values in pytorch (Paszke et al., 2017). The average accuracy over 8 random states is 63.19% for T-CNN. Using a CNN with more parameters, a higher accuracy is achievable. The smaller network is selected to see the impact of delayed feedback.

#### 4.1.2 T-CNN AND BASIC CL-CNN RESULTS

Table 1 illustrates the results obtained using various queue sizes over 8 different initial random states to initialize neural network weights. The second row in table shows the T-CNN accuracy. Other rows illustrates continual learning with delayed feedback accuracy(CL-CNN accuracy) with various queue sizes. All the accuracies are reported in percentage(%).

Table 1: Accuracy against various queue Sizes and random states

| Queue Size | 1 | 2 | 3 | 4 | 5 | 6 | 7 | 8 | Mean |
|---|---|---|---|---|---|---|---|---|---|
| - | 62.79 | 65.04 | 61.96 | 63.42 | 63.62 | 62.25 | 62.98 | 63.43 | 63.19 |
| 100 | 61.36 | 62.39 | 60.38 | 61.06 | 61.25 | 59.26 | 60.02 | 62.01 | 60.97 |
| 200 | 61.84 | 62.47 | 61.13 | 62.24 | 62.18 | 61.02 | 60.94 | 61.91 | 61.72 |
| 500 | 62.92 | 64.45 | 62.67 | 63.75 | 62.88 | 62.17 | 62.47 | 63.57 | 63.11 |
| 1000 | 64.96 | 66.16 | 63.40 | 65.25 | 66.06 | 64.20 | 64.24 | 65.62 | **64.99** |
| 2000 | 68.52 | 70.27 | 67.96 | 69.52 | 69.65 | 68.01 | 68.24 | 70.08 | **69.03** |

The accuracy of CL-CNN is steadily increasing with queue size. Having a bigger memory to hold unknowns will help to make good decisions. With $10,000$ samples for **growing** in CIFAR10 , increment of the queue size is stopped at $2,000$. When queue sizes are $1000, 2000$, the accuracy is above T-CNN accuracy.

Table 2: Accuracy without final queue of samples vs final accuracy

| Queue Size ($\lambda$) | Accuracy for samples ($10000 - \lambda$) | Delayed Feedback accuracy($10000$) |
|---|---|---|
| 100 | 61.27 | 60.97 |
| 200 | 62.40 | 61.72 |
| 500 | 64.91 | 63.11 |
| 1000 | 68.70 | 64.99 |
| 2000 | 77.39 | 69.03 |

The table 2 shows an interesting result of CL-CNN. The results are averaged over 8 random states which are used to initialize network weights. The final queue contains the hardest examples for CL-CNN. For instance, if the queue size is 100, then 9900 samples are processed first. Finally, the last content of the queue is processed. Infact, the samples which get high confidences will get a higher chance to leave the queue. These queues can be used to assist clustering the samples into harder and simpler samples for the given model.

### 4.1.3 COLOR FEATURES WITH DELAYED FEEDBACK

Table 3 shows the results when the feedback is given with a reference image as discussed in section 3.2. This reference image is picked from the images used for **seeding**. For each class 100 reference images are selected. For every feedback a random image out of the relevant class is picked. Using the reference image, the color similarity is calculated and the updates are done based on figure 3. It is quite evident that giving more information with feedback resulted in increase in accuracy with all different queue sizes of CL-CNN. The final accuracy has increased from $69.03\%$ to $73.26\%$. The initial layers of CNN function as modules which can detect color and basic shapes. But adding color as feedback in basic CL-CNN improves over T-CNN with a larger margin $10\%$ as well as basic CL-CNN without color features with a margin of $4\%$. Color features in feedback play a different role in improving accuracy rather than initial layers of CNN which detect colors.

Table 3: Accuracy with color features vs accuracy without color features

| Queue Size | Accuracy with color features (%) | Accuracy without Color (%) |
|---|---|---|
| 100 | 62.73 | 60.97 |
| 200 | 63.62 | 61.72 |
| 500 | 65.56 | 63.11 |
| 1000 | 68.34 | 64.99 |
| 2000 | 73.26 | 69.03 |

### 4.1.4 BATCH UPDATE WITH DELAYED FEEDBACK

Table 4 shows the results with batch update. Adding batch update to the feedback with color decreases the accuracy over CL-CNN with color feedback. But it ended up in a faster training time(nearly 3 time faster for the batch size of 4). A batch of 4 is used to update. Sometimes getting 4 samples which can meet the updating rule is not possible, in those scenarios model is updated with a smaller batch or even no update. Rather than a random feedback, color feedback with batch update improves a bit as shown in the table.

Table 4: Accuracy with color feedback and batch update vs accuracy of basic CL-CNN

| Queue Size | Accuracy with color and batch update (%) | Basic Delayed Feedback(%) |
|---|---|---|
| 100 | 61.92 | 60.97 |
| 200 | 62.60 | 61.72 |
| 500 | 63.98 | 63.11 |
| 1000 | 65.67 | 64.99 |
| 2000 | 69.35 | 69.03 |

## 4.2 DECISION TREE BASED EXPERIMENTS

The framework is deployed in decision trees. Small changes are done to the framework. Instead of maintaining a queue, DTs are updated instantly. Unlike neural network, partial update to the model for a sample is not possible. Hence, for each new sample, the whole dataset is fitted. In addition, if the confidence values are higher, no update is done. The model is updated if the confidence value is in $[0.5, \epsilon]$.

Digits data in scikit learn (Pedregosa et al., 2011) is used to train various models.(It contains 1797 digit images(0-9) of size $8 \times 8$). Various sizes of decision tree depth are checked. Higher depths are not used here to avoid overfitting in the **seeding** phase. T-DT denotes a Traditional Decision Tree and CL-DT denotes a Continually learnt Decision Tree.

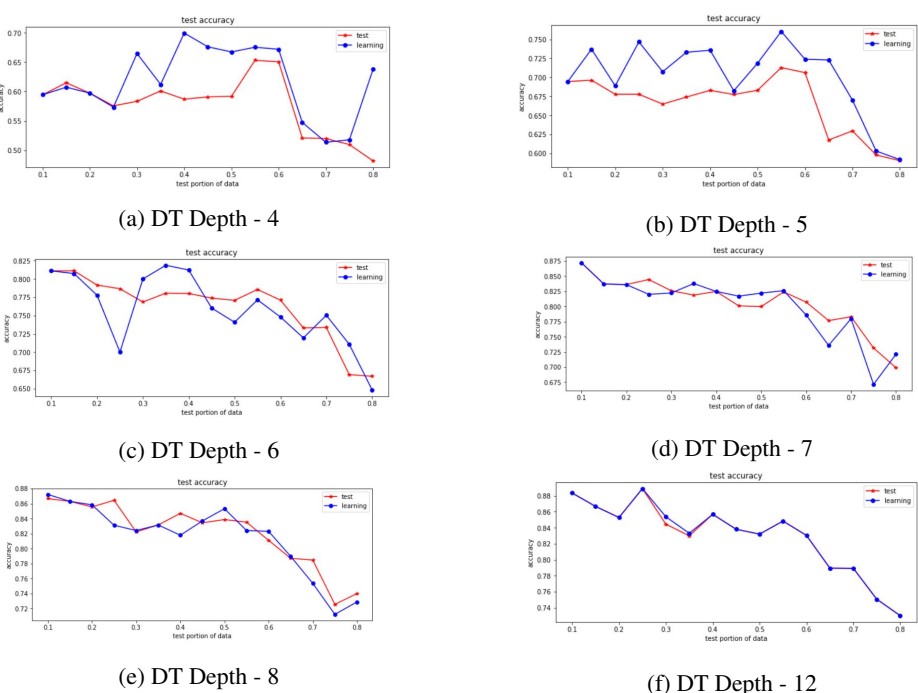

(a) DT Depth - 4

(b) DT Depth - 5

(c) DT Depth - 6

(d) DT Depth - 7

(e) DT Depth - 8

(f) DT Depth - 12

Figure 4: Accuracy vs Seeding-Learning Split

Test corresponds to the test accuracy of a T-DT model whereas learning corresponds to the **growing** accuracy of a CL-DT. Figure 4 illustrates that when DT depth is small, the model performs better than T-DT. Depth 4 and 5 of CL-DT produce better results than a T-DT in most of the different evaluating portion of the data. With increase in depth, the difference in accuracy is not significant. The reason is that model ovefits the data in training and seeding phases. This leads towards 100% seeding and training phase accuracy. Hence, for the new data sample, the DT gives high confident values for the decisions. As per the update rule, CL-DT is updated if confident values are not that strong($Y$ is the confidence vector such that $\forall y \in Y, y \notin (0.8, 1]$). Hence, in the case high depth

CL-DTs, actually no update is really happening per our rule (The reason is that we update the model if the confidence values / probability values fall in a range $[0.5, 0.8]$. If the confidence values are above $0.8$, then the number of updates would go towards $0$ with the height of CL-DT.). Hence, both CL-DT and T-DT accuracies are around the same.

The number of samples is small. Instead of having a bigger queue, the updates are done on the model instantly (technically there is no queue. But this can be changed if the number of samples are higher. However, the update of the model will take longer time to train). Similar to neural network, online update is not possible with sci-kit learn DT models (D3, C4.5, C5.0 and CART). The updates are done for the whole sample set.

## 5 CONCLUSION AND FUTURE WORKS

Model-agnostic continual learning framework is proposed and deployed in CNNs and DTs. The results show improvement in classification accuracy using a delayed feedback system. The framework works well with the machine learning models which can be learnt online. CL-CNN is a clear example where images can be learnt continually. However, continual learning differs from online learning in the way of updating weights and learning rates. The nature of DT and SVM does not match the framework since everything is updated together in these models(Batch Learning). Similar to DT similar framework version can be used to SVM, but the models have the limitations of scaling if the dataset used to learn blows up. Buffering recent data in a queue gives the model a chance to select an easier sample to learn. Increase in queue size resulted in better accuracy which says that having a larger buffer data helps.

There are few limitations with this framework. Switching testing data of CIFAR 10 for seeding and training data for growing data resulted in accuracies which are around $55\%$. CL-CNN improved it by mere $1\%$ with the queue size of $5000$. There is a diminishing return with the improvement of accuracy when growing data surpassed the seeding data by a bigger factor. This is a limitation with this framework.

Continual Learning framework discussed in this work can be used with supervised learning. The difference from a T-CNN(with Stochastic Gradient Descent(SGD)) is that the learning rate varies with the confidence values whereas in T-CNN(with SGD) the learning rate is reduced over number of epochs and usually fixed during an epoch. Similar to ADAM optimizer ((Kingma & Ba, 2014)), proposed CL framework can be used as an adaptive learning technique where the learning rate is a function of confidence values.

Delayed Feedback is given with a label and a reference image where one is an output element and other one is the input element. The update is not done if the similarity is not high. There are chances where conventional color histograms, shapes and textures differ by a good margin, but still the objects belong to the same category. More sophisticated ways are needed to update with intrinsic input properties. In addition, instead of a label and a reference image, more auxiliary information such as text can be given as feedback. Feedback is any form of information which can be used to do learning.

This work creates further directions in research areas of continual learning, and delayed feedback systems.

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
