# OpenReview forum: "Continual Learning with Delayed Feedback"
_ICLR.cc/2020/Conference — Reject_

### Official Review · AnonReviewer3 · 2019-10-21
**Official Blind Review #3**

**Rating:** 1

**Review:**

This paper describes a method that draws inspiration from neuroscience and aims to handle delayed feedback in continual learning (ie. When labels are provided for images after a phase of unsupervised learning on the same classes). It is an interesting idea, and worth exploring.

I found the paper quite hard to follow at times, and I suggest the authors go through the paper in detail to address some of the issues with grammar and clarity of explanation.

In addition, I think the paper is lacking some grounding and context in terms of what problem is being solved and what previous work exists.
For example, based on the experimental setup section it seems like the problem being addressed is that of unsupervised learning on CIFAR images followed by supervised learning on images with labels (IE. Delayed feedback) - is this the case? If so, this is much more a semi-supervised learning or fine-tuning problem than a continual learning problem (which typically looks at a single class at a time or some other non-stationary sequence of tasks). Either way, the recent literature in semi-supervised learning and continual learning should be referenced - see the citations below as a few examples, and consider referencing and more closely perusing some of the examples in the cited review paper by Parisi et al (2019).

Lastly, the experiments show the performance as a function of queue length for different features and for CNNs versus decision trees, but there is no comparison to existing methods and very simple models are used - this means that again, it's difficult to gauge the efficacy of the approach and place this in the context of prior art.

Unfortunately, I think the paper in its current state does not meet the bar for ICLR - I suggest the authors consult the vast literature in semi-supervised and continual learning, and try to place their work in this context, along with external comparisons.


Nguyen, Cuong V., et al. "Variational continual learning." arXiv preprint arXiv:1710.10628 (2017).

Lopez-Paz, David, and Marc'Aurelio Ranzato. "Gradient episodic memory for continual learning." Advances in Neural Information Processing Systems. 2017.

Miyato, Takeru, et al. "Virtual adversarial training: a regularization method for supervised and semi-supervised learning." IEEE transactions on pattern analysis and machine intelligence 41.8 (2018): 1979-1993.

**Experience Assessment:**

I have published one or two papers in this area.

**Review Assessment: Checking Correctness Of Derivations And Theory:**

I carefully checked the derivations and theory.

**Review Assessment: Checking Correctness Of Experiments:**

I assessed the sensibility of the experiments.

**Review Assessment: Thoroughness In Paper Reading:**

I read the paper at least twice and used my best judgement in assessing the paper.

---

### Official Review · AnonReviewer2 · 2019-10-24
**Official Blind Review #2**

**Rating:** 1

**Review:**

The paper claims to tackle a semi-supervised continual learning problem where the feedback or the labeled data is delayed and is provided based on the model performance. Authors do not provide standard benchmarks for comparison and no baseline is considered.

The idea of using unlabeled data for continual learning is interesting and to the best of my knowledge this is the first work that suggests using delayed feedback for continual learning but unfortunately they do not consider measuring forgetting and this work seems an online learning method using delayed feedback.

I vote for rejecting this paper due to the following reasons. I have listed the issues in chronological order and not their importance.

1- I start with the writing. The paper, in its current form, needs to be thoroughly proofread and reorganized. The text does not read well and is vague in most parts (for example section 3 and 4). The text is informal in some parts (ex. in Figure 1). There are also grammar errors and typos for which I have found passing my writing through the free version of Grammarly very helpful in getting rid of most such errors.

2- As one of the main motivations for the paper, authors claim humans learn continuously in an unsupervised fashion (paragraph one). I disagree with this statement because we all have been constantly learning from the feedback we have been receiving the environment throughout our lives. For example we all have learned how to walk by falling on the ground multiple times and using the pain signal in our muscles as a negative feedback to correct our movements. Getting corrected while speaking or question answering in our dialogues are examples of receiving feedback from the environment letting our learning behavior receiving lots of supervision.

3- The related work section misses significant number of prior work on continual learning (I have provided a short list at the end [4,5,6] but authors are strongly encouraged to read more on this literature). However, my biggest concern is that I this work should not be introduced as a continual learning algorithm. The proposed method is an online learning method with delayed feedback which has been extensively studied before. Authors should consider citing the pioneering work in this field such as Weinberger & Ordentlich (2002) [2] or Joulani et al from ICML 2013 [3]. Providing comparison to [3] is strongly encouraged. Also note that the citation for “catastrophic forgetting” is wrong and should be corrected to McCloskey & Cohen (1989).

4- The figures and tables do not meet the conventional scientific standards and have to be significantly improved.

5- Authors use softmax probabilities as a confidence score which are known to be uncalibrated by large as deep models are usually overconfident about their predictions. (see [1] for example). Was this investigated at all? Using a calibration technique might be able to help with this [1].

6- On page 4, paragraph 5, the authors claim that “in continual learning instant update is not done”. This is vague to me and I think it is not true because there are plenty of supervised continual learning approaches where the labeled data is available when a task is learned (for example [4,5,6])

7- The experimental setting is not well designed and does not use a standard continual learning setting and there is no baseline included which are very important reasons for rejecting this paper. Authors can benefit from applying their method on standard benchmark datasets commonly used in the literature to provide a fair comparison. Most importantly authors should evaluate their method against prior work.

8- Exploring continual learning for decision trees is completely vague not justified in the paper.

[1] Guo, Chuan, et al. "On calibration of modern neural networks." Proceedings of the 34th International Conference on Machine Learning-Volume 70. JMLR. org, 2017.
[2] Joulani, Pooria, Andras Gyorgy, and Csaba Szepesvári. "Online learning under delayed feedback." International Conference on Machine Learning. 2013.
[3] Weinberger, Marcelo J., and Erik Ordentlich. "On delayed prediction of individual sequences." IEEE Transactions on Information Theory 48.7 (2002): 1959-1976.
[4] Kirkpatrick, James, et al. "Overcoming catastrophic forgetting in neural networks." Proceedings of the national academy of sciences 114.13 (2017): 3521-3526.
[5]Lopez-Paz, David, and Marc'Aurelio Ranzato. "Gradient episodic memory for continual learning." Advances in Neural Information Processing Systems. 2017.
[6] Serrà, J., Surís, D., Miron, M. & Karatzoglou, A.. (2018). Overcoming Catastrophic Forgetting with Hard Attention to the Task. Proceedings of the 35th International Conference on Machine Learning, in PMLR 80:4548-4557

**Experience Assessment:**

I have published one or two papers in this area.

**Review Assessment: Checking Correctness Of Derivations And Theory:**

N/A

**Review Assessment: Checking Correctness Of Experiments:**

I carefully checked the experiments.

**Review Assessment: Thoroughness In Paper Reading:**

I read the paper thoroughly.

---

### Official Review · AnonReviewer1 · 2019-10-28
**Official Blind Review #1**

**Rating:** 1

**Review:**

General

The paper is quite hard to follow. The figures are very coarse and the problem definition is not given very well. The paper claims that it is about continual learning, but it does not give ANY experimental results on the continual learning benchmarks. The paper seems to be dealing with an online learning with unlabeled data.

Con & Questions:

- Fig 3 shows the update rule, but there is no explanation on I_ref or X_1.
- The paper says it generates random one-hot vector when queue is full, but what does it have to do with delayed feedback?
The algorithm requires a completely trained model and a queue that needs to store large amount of data. Then, what is the good nature of this method?
There is no baseline in the experimental results.
The T-CNN result on Cifar-10 is too low. This makes the result dubious.

**Experience Assessment:**

I have published one or two papers in this area.

**Review Assessment: Checking Correctness Of Derivations And Theory:**

I assessed the sensibility of the derivations and theory.

**Review Assessment: Checking Correctness Of Experiments:**

I assessed the sensibility of the experiments.

**Review Assessment: Thoroughness In Paper Reading:**

I read the paper at least twice and used my best judgement in assessing the paper.

---

### Decision · Program_Chairs · 2019-12-19

**Decision:**

Reject

**Comment:**

This paper claims to present a model-agnostic continual learning framework which uses a queue to work with delayed feedback. All reviewers agree that the paper is difficult to follow. I also have a difficult time reading the paper.

In addition, all reviewers mentioned there is no baseline in the experiments, which makes it difficult to empirically analyze the strengths and weaknesses of the proposed model. R2 and R3 also have some concerns regarding the motivation and claim made in the paper, especially in relation to previous work in this area.

The authors did not respond to any of the concerns raised by the reviewers. It is very clear that the paper is not ready for publication at a venue such as ICLR at the current state, so I recommend rejecting the paper.